# Review of Experimental Methods for Measuring the Ignition and Combustion Characteristics of Metal Nanoparticles

**DOI:** 10.3390/nano10102008

**Published:** 2020-10-12

**Authors:** Vladimir Zarko, Anatoly Glazunov

**Affiliations:** 1Voevodsky Institute of Chemical Kinetics and Combustion, Russian Academy of Sciences, 630090 Novosibirsk, Russia; 2Laboratory for Designing Elements of Rocket and Space Technology, Tomsk State University, 634050 Tomsk, Russia; gla@niipmm.tsu.ru

**Keywords:** ignition, metal, nanoparticles, combustion mode, heat transfer, free-molecular, burning time

## Abstract

Investigations in recent decades have shown that the combustion mechanism of metal particles changes dramatically with diminishing size. Consequently, theoretical description of the ignition and combustion of metal nanoparticles requires additional research. At the same time, to substantiate theoretical models, it is necessary to obtain objective experimental information about characteristics of ignition and combustion processes, which is associated with solving serious technical problems. The presented review analyzes specific features of existing experimental methods implied for studying ignition and combustion of metal nanoparticles. This particularly concerns the methods for correct determination of nanoparticles size, correct description of their heat-exchange parameters, and determining the ignition delay and combustion times. It is stressed that the problem exists of adequate comparison of the data obtained with the use of different techniques of particles’ injection into a hot gas zone and the use of different methods of reaction time measurement. Additionally, available in the literature, data are obtained for particles of different material purity and different state of oxide layer. Obviously, it is necessary to characterize in detail all relevant parameters of a particle’s material and measurement techniques. It is also necessary to continue developing advanced approaches for obtaining narrow fractions of nanoparticles and for detailed recording of dynamic particles’ behavior in a hot gas environment.

## 1. Introduction

Metal nanoparticles are known from very distant times; for example, the famous Cup of Lycurgus, colored stained-glass windows in churches, etc. It can be mentioned that scientific studying the properties of nanoparticles (gold and silver) started in the 19th century in fundamental work of M. Faraday [1]. However, intensive research in this area began relatively recently, and this happened due to the development of methods for obtaining nanoparticles by evaporation–condensation [2] and electric wire explosion [3].

The history of metal nanoparticles application in the combustion and explosion processes developed according to a typical scenario for new discoveries: first, a rapid euphoria and great expectations, then a decline in interest, a decrease in activity, followed by a growth of interest and the achievement of real positive results. This situation was described by M. Zachariah in an editorial note published in 2011 [4]. At present, metal nanoparticles are widely used as modifiers of the burning rate of solid propellants, components of pyrotechnic compositions, etc.

It can be noted that nanoparticles are intensively studied as individual objects, the products of production by various technical methods, and also as the combustion products of original macro- and micro-particles of metals. In the first case, there are questions of atomization (dispersion) of individual particles, since aggregation processes at nanoparticles level proceed very fast, and it is necessary to have evidence that the procedure is carried out with a given particle size. Secondly, one has to consider general picture of the metal particles combustion when changing their characteristic size. Research in recent decades has shown that the combustion mechanism of metal particles changes dramatically along with changing their size. This is clearly demonstrated by the diagrams of the burning time dependency on the particle size, Figure 1, Figure 2 and Figure 3.

A sort of generalized information on the burning time t_b_ dependency of the Al and B particles on size [13] is presented in Figure 4, which is generally true for other metals [11]. Here, one can see that for particles with a size greater than 20–30 microns, a quadratic dependency in the form of t_b_~D^2^ takes place, then with a decrease in size, the dependency on the particle size weakens and takes the form t_b_~D^1^, while for submicron particles, it becomes t_b_~D^n^, where n = 0.3–0.5. The main reason for changing the type of dependency for the burning time is a change in the mechanism of the particle heat exchange with the gas environment [14,15,16].

Thus, theoretical description of the processes of ignition and combustion of metal nanoparticles requires additional research. However, to substantiate the theoretical models, it is necessary to obtain objective experimental information about the characteristics of these processes, which is associated with solving serious technical problems. The purpose of the present article is to analyze specific features of experimental methods for studying and recording the characteristics of the ignition and combustion of metal nanoparticles. In particular, this concerns the methods for correct determining the size of nanoparticles, as well as correct accounting for their heat-exchange parameters and recording the ignition delay and combustion times.

## 2. Combustion Mechanism Variation with Metal Particle Size

The equation shows the qualitative assessment of the prevailing reaction mechanism of a metal particle that one can make via calculation of the Damkohler number *D_a_* representing the ratio of diffusion and chemistry time scales:*D_a_*~*D_p_*/*D_ox_*(1)

Here, *D* is the particle diameter, *p* the gas pressure, and *D_ox_* the oxidizer diffusion coefficient.

At *D_a_* >> 1, a diffusion-controlled process of metal particle oxidation is implemented, whereas at *D_a_* << 1, the oxidation is governed by chemical kinetics. Equation (1) indicates that the oxidation is characterized by a diffusion-controlled process in the case of large particles and high pressure, while in the case of small particles and low pressures, it is characterized by a kinetically controlled process. Approximate estimates for Equation (1) show [17] that for boron particles (according to [7]), the diffusion-controlled process is realized at sizes over 75 microns at 1 atm and, for aluminum particles (according to [18]), at sizes over 100 microns at 1 atm. Note that these limits decrease in direct proportion to the medium pressure. At the same time, it is important to note that the above estimates are very approximate and more accurate analysis of the heat-exchange modes can be made on the basis of the Knudsen number, which is defined as the ratio of the free molecules path length *λ* and the particle size *D*/2: *Kn* = 2*λ*/*D*.

It can be considered as two limiting cases: *Kn* >> 1 and *Kn* << 1 [14]. In the first case, the particle diameter is less than the free gas molecules path length, and this mode of heat exchange is called free-molecular. In the second case, the particle diameter significantly exceeds the free path length, and the heat exchange occurs in the continuous medium mode. It is assumed that the first case corresponds to the values of *Kn* > 10, and the second, when *Kn* < 0.01. At intermediate values 10 > *Kn* > 0.01, a transient heat-exchange mode is implemented. The Knudsen number is calculated using Equation (2):(2)$$Kn=\frac{RT}{\sqrt{2}\pi D_{a}^{2}N_{A}pD},$$
where *R* is the universal gas constant, *T* the temperature, *D_a_* the diameter of the ambient gas molecule, *N_A_* the Avogadro’s number, and *p* the pressure. The particle sizes corresponding to the Knudsen numbers 0.01 and 10 are represented as functions of pressure at temperatures of 300 and 3000 K in Figure 5. It can be seen that at a pressure of 1 atm and a gas temperature of 3000 K, the particle size corresponding to the assumption of a continuous medium is equal to 70 microns. It is reduced by 10 times if the pressure increases to 10 atm or when the temperature falls down to 300 K.

It is obvious that during the evolution of the size of burning metal particles, the mechanisms of heat exchange with the gas medium change (radiation heat exchange must be considered additionally) and the calculation of the total burning time must be made, accounting the stages. An illustration [19] of the variation of combustion mechanisms versus the aluminum particle size is shown in Figure 6.

Ten micron and larger particles are characterized by a structure with detached diffusion flame and maximum temperature close to the adiabatic flame temperature of Al—the air system. For submicron particles, the reactions practically proceed on the surface of the particles. The temperature of flame is almost equal to the surrounding gas temperature, especially at low pressures. It is not exceeding the aluminum boiling point and decreases when going far from the surface. In the case of intermediate size particles, the flame approaches the surface of a particle and the maximum temperature can be significantly lower of adiabatic one.

In general, the qualitative picture of the burning metal particle depends on the ratio of the metal boiling point and the boiling—dissociation temperature of its oxide. The concept of “Glassman’s criterion” [20,21,22] is known in the literature, which states that if the metal boiling point is lower than its oxide boiling point-dissociation temperature, combustion occurs in the vapor phase. With the reverse ratio of the characteristic temperatures, heterogeneous combustion occurs on the surface of the particles. According to the literature data, the metals Be, Cr, Al, Fe, Hf, Mg, Li, and Ti should burn in pure O_2_ at atmospheric pressure in the vapor phase mode forming a diffusion flame. At the same time, Si and Zr should burn heterogeneously [23].

In the case of metalloid B, the situation is special: due to abnormally low boiling point its oxide vaporizes on the expense of heat feedback from the flame, but the energy is not sufficient to reach the metal boiling point, and potentially possible heterogeneous combustion mechanism is not realized. It is important to emphasize that the necessary condition for the existence of a vapor-phase mechanism of metal combustion is the excess of the flame temperature over the metal boiling point. Therefore, if there exists a noticeable heat loss, the combustion mode of metals with a relatively small difference in the boiling temperatures of metal and oxide (within 400 K; Cr, Fe, Hf, Ti) can be changed from vapor-phase to heterogeneous. Similarly, the combustion mode changes when the composition of the oxidizer changes: aluminum combustion occurs heterogeneously with CO_2_ and H_2_O vapors (diminished flame temperature). It should be noted that these arguments are qualitative in nature and are valid for aluminum mainly at moderate pressures and temperatures. Particularly, the combustion mode with evaporation can be realized for small particles of aluminum at high pressure and gas temperature.

Thus, in general, the total burning time $\tau_{b}^{\sum }$ of a relatively large size metal particle must be calculated taking into account the changes of heat-exchange modes (from continuum through transition to free-molecular regimes):(3)$$\tau_{b}^{\sum }={\tau_{b}^{\mathrm{cont}}|}_{D_{0}}^{D_{1}}+{\tau_{b}^{\mathrm{trans}}|}_{D_{1}}^{D_{2}}+{\tau_{b}^{\mathrm{free}}|}_{D_{2}}^{0}$$

Here, *D*_0_ is the initial particle diameter, *D*_1_ the size corresponding to the value of the Knudsen parameter *Kn* = 0.01, and *D*_2_ the size corresponding to the value *Kn* = 10.0. It should be noted that, in available literature, this approach to the calculation of the total burning time of metal particles to date has not been implemented, and in published works, it was assumed by default that the selected burning mechanism is valid from the beginning to the end of a particle’s combustion. To the authors’ knowledge, there is only one work on the combustion of boron particles, where the attempt to develop such an approach has been made [24].

## 3. Correct Determination of Nanoparticles Size

Determination of nanoparticles’ size is rather difficult technical problem, which also involves the issue of disconnecting the aggregated and sintered particles. In many cases, the available results of such measurements do not look as well-substantiated. In research conducted with Al nanoparticles in the shock wave tube [25], the particles were injected radially into the tube using a pneumatically driven piston. High resolution scanning electron microscope (SEM) was employed to measure for each sample the particle size distribution, and more than 100 diameter measurements from each sample were usually made for obtaining particle size distribution.

Then, the particles were characterized by number average and mass average diameters. In particular, a fraction called “50 nm SkySpring NanoMaterials” had the number average and mass average diameters 73.2 nm and 80.9 nm, correspondingly. In [25], there is no explanation of how the particles probes for the SEM analysis were prepared in order to represent total size distribution and why only about 100 particles were measured. An example of particle size distribution for 50 nm sample is shown in Figure 7. It can be recognized that there are particles, which size is approximately twice larger than the medium one. Consequently, when comparing the results corresponding to different samples, it is necessary to take account of the presence of the highest size particles. The wide distribution of particle sizes might be a reason for weak dependency of burning time on the nanoparticles’ size. Another reason could be different origin and properties of particles from different sources: mass content of Al and oxide-coating thickness.

One more reason for uncertainty in determination of the nanoparticles’ ignition and burning times can be aggregation and agglomeration of original particles leading to formation of bigger size particles with longer operation time. In [25], the authors discuss this issue and state that in conditions of shock waves the strong shear forces could be effective at breaking weak agglomeration in nanoparticles. Unfortunately, the evidences of effective disaggregation of particles are not presented. At the same time, there are known cases of studying the reactive characteristics of metal nanoparticles in laminar flames with the injection of particles in a high temperature zone, where the particles’ agglomeration may have significant effect. As an example, Figure 8 presents a setup used in [26] for studying isolated boron particles ignition and combustion in fluorine-containing environments. The boron particles were injected in a spatially uniform post-flame region from a fluidized-bed particle feeder. The particles mixture in a bed consisted of boron (1–3 μm) and large silica particles (70–150 μm in diameter) at 1:20 average mass ratio. Coarse silica particles are assumed to break up existing boron particle agglomerates. No special analysis has been made to determine real particles size in the flame. A similar approach for feeding the boron particles of 2.3 μm average size was used in [27].

A detailed examination of metallized particles ignition and combustion behavior was conducted in the beginning stage of such research in [28]. The particles were characterized by sizes provided by manufacturers; they included nanoAl in the range of sizes from 24 to 450 nm and micron-sized Al and Ti: Al, TiH_2_, Ta, Hf, and B particles. Two types of particles introduction into the flame were employed: with use of commercial Meinhard™ nebulizer and a homemade pneumatic system. In the last case, the fuel sample powder was placed between two aluminum foil burst disks and injected shortly into the hot zone of steam generator after the disks bursting. In the first case, the nebulizer generated liquid droplets of average size of 20 microns that provided an average weight of aluminum in a drop equal to that in an approximately 2 micron diameter particle. The sizes of individual (non-aggregated) particles remained unknown. In the case of the pneumatic feeder, the dispersibility of ignited and burning particles was also unknown. All data on reactive characteristics were discussed in terms of effective sizes of original particles. The authors of [28] reported about possible errors in the results of research due to wide particle size distributions of aluminum samples and the agglomeration effect and mentioned that it is nearly impossible to entirely disperse nanosized particles.

During past two decades, there were undertaken several attempts by different groups to elaborate effective methods of obtaining narrow size particle distribution samples of Me nanoparticles and methods of in situ measuring size of individual particles. Some positive and promising results were obtained in this direction by the group of Prof. M.R. Zachariah at the University of Maryland.

In the paper devoted to kinetic measurements of size-resolved aluminum nanoparticles’ oxidation, it was formulated [29] that the intrinsic reactivity of nanoparticles would be better to explore in the absence of other rate-limiting kinetic effects associated with the heat and mass transfer. For this end, it is necessary to produce isolated particles and expose them to reactants in conditions of controlled environment with simultaneous measurement of size and residence time. Individual aluminum nanoparticles were generated by means of DC arc discharge or laser ablation (Figure 9) that allowed the authors to obtain free-of-oxygen particles.

The particles were oxidized at different temperatures (up to 1100 °C) in an aerosol flow reactor. To measure the particles’ elemental composition and size as a function of the furnace temperature, the aerosols after reactor were sampled into the single particle mass spectrometer (SPMS). It is composed of an aerodynamic lens inlet, three-stage differential vacuum system, free-firing pulsed laser for ionization of a particle, and linear time-of-flight mass spectrometer. This provides an efficient simultaneous measurement of individual particles composition and size in terms of metal conversion into oxide extent. In front of the SPMS, it was installed a differential mobility analyzer (DMA) for sending monodisperse aerosols directly in the SPMS in order to derive the particle size and ion signal intensity relationship. Additionally, to measure particle size distribution, the DMA was coupled with a condensation particle counter. Finally, to explore the morphology of aluminum particles, a transmission electron microscopy (TEM) was employed. The size of individual particles selected by DMA (particle mobility size) was also measured using a time-of-flight mass spectrometer and compared with real size of sampled particles measured by TEM. It was revealed that aerosol particles (mobility size) consist of about 30 primary particles, and all kinetic data were referred to mobile size, which was determined at varying flow reactor conditions. Nevertheless, it can be concluded that in this case researchers had deal with the narrowest particle size distributions and obtained some very important information about the reactive characteristics of metal nanoparticles.

In particular, it was established in [29] that when primary and mobility particle sizes decrease, the reactivity of aluminum nanoparticles increases. It was also found that activation energy of the aluminum oxidation decreases when particle size decreases. Another important finding was that there exists significant difference of SPMS results with those obtained with classical non-isothermal gravimetric measurements method (TGA-ThermoGravimetric Analysis). Opposite to the onset temperature for aluminum nanoparticle oxidation below the melting point (510–530 °C, TGA), the SPMS indicates that oxidation starts just above the aluminum melting point. Besides, measured with the SPMS oxidation reaction rates became significantly different from those that were obtained from the conventional TGA procedures. All these differences are attributed to inherent features of bulk thermal methods application that resulted from ensemble effects associated with the relatively large mass of bulk samples.

Interesting results regarding combustion mechanism of metal nanoparticles were obtained with SPMS application in [30,31] when having deal with study of Ti and Zr particles. Due to high boiling points (Ti: 3560 K, Zr: 4650 K), which exceed the volatilization temperatures of corresponding oxides, the processes of Ti and Zr combustion are mainly dominated by surface reactions (Glassman’s criterion). Laser ablation was used to generate nanoparticles of Ti and Zr in an inert environment and then the particles were introduced into the post-flame zone of a laminar CH_4_/O_2_/N_2_ diffusion flame with the temperature from 1700 K to 2500 K. A modified DMA device employed as a size selection tool was combined with a high-speed camera for registration of combustion characteristics. In addition, the metal particles were sampled by a nanometer aerosol sampler at different heights above the burner and characterized by TEM. Analysis of sampled “fresh” Ti particles revealed that they consist of aggregates composed of primary particles with size of 10.3 ± 0.4 nm. After passing the high temperature burner zone, the resulting particle morphology changed from aggregate to isolated sphere. Similar behavior demonstrated particles of Zr. The measured burning times were plotted at diagrams for particles size measured by DMA (mobile size) or estimated after sintering (TEM size). An example of such diagrams for Ti particles is presented in Figure 10.

It is seen that the power law exponent for burning time vs. particle size dependency equals 0.62 for DMA size and 0.89 for sintering size. These values are higher than typical values (ca. 0.3–0.5) for metal nanoparticles and for micron-size Ti and Zr particles [32]. Corresponding values of exponent for Zr particles comprise 0.53 and 0.77, correspondingly.

The visual observations of Ti and Zr nano particles combustion demonstrated very short streaks of the particle aggregates. This allows us to assume that their combustion proceeds in a practically isothermal environment. Using the measured oxidation zone temperature profiles, the temperature difference on an average streak length was estimated to be about 20 K. Therefore, isoconversion techniques were used to obtain apparent Arrhenius parameters for the nanoparticles under study oxidation reaction. The estimations were performed for chosen mobile particle size (145.9 nm) via analysis of the streaks recorded at various flame conditions. They were determined for Ti pre-exponential factor of 7.5 × 10^5^ s^–1^ and activation energy of 56 kJ/mol and, for Zr, 3.4 × 10^5^ s^–1^ and 43 kJ/mol, respectively. Actually, these results became available due to essential advancements in developing techniques, which allowed us to obtain “monodispersal” aerosols of metal nanoparticles.

## 4. Correct Determination of Energy Accommodation Coefficient

As is mentioned in Section 2, the conduction heat transfer for nanoparticles obeys the free-molecular regime. It was underlined in [25] and other works that the poor efficiency of this type of heat transfer is caused by the low value of energy accommodation coefficient α. This coefficient characterizes the behavior of gas molecules in their collisions with a solid or liquid body surface. It is calculated as the ratio of the gas molecule energy transferred during a collision to the theoretical maximum value under complete energy accommodation:$$a=\left(E_{0}-E_{1}\right)/\left(E_{0}-E_{s}\right),$$
where *E*_0_ and *E*_1_ are the average energies of incident and scattered gas molecules, respectively, and *E_s_* the average energy of gas molecules in thermal equilibrium with the surface. The accommodation coefficient value depends on the nature of particle surface and its state as well as on the gas mixture composition and pressure.

The correct knowledge of accommodation coefficient is important for objective description of nanoparticles heat transfer that may help to substantiate possible mechanism of particles’ reacting. The particle energy conservation equation can be written as
$$m_{p}C_{p}\frac{dT}{dt}={\dot{Q}}_{gen}-{\dot{Q}}_{rad}-{\dot{Q}}_{cond},$$
where *m_p_* is the mass of particle.

The chemical heat generation rate is described by the formula
$${\dot{Q}}_{gen}=A_{p}\phi N_{ox}cq/4,$$
where *A_p_* is the particle surface area, *ϕ* the sticking probability, *N_ox_* the oxidizer molecules number concentration, *c* the molecular velocity, and *q* the reaction heat. The sticking probability characterizes the fraction of collisions that results in chemical reactions.

Radiation heat transfer is described by the formula
$${\dot{Q}}_{rad}=\varepsilon_{p}\sigma A_{p}\left(T_{p}^{4}-T_{a}^{4}\right),$$
where *ε_p_* is the emissivity of the particle.

The free-molecular heat conduction regime is described by the formula
$${\dot{Q}}_{cond}=\alpha \pi D^{2}\frac{p_{a}\sqrt{8k_{B}T_{a}/\pi m_{a}}}{8}\left(\frac{\gamma +1}{\gamma -1}\right)\left(\frac{T_{p}}{T_{a}}-1\right),$$
where *α* is the energy accommodation coefficient, *k_B_* the Boltzmann constant, *m_a_* the average mass of the gas molecule, and *γ* the ratio of specific heats. The subscripts *a* and *p* stand for the ambient gas and particle, correspondingly.

The attempts to describe theoretically the interaction of gas and surface atoms were undertaken in the past in several works [33,34,35] and later the estimate for the upper bound for the energy accommodation coefficient was presented in [36]. The principle of detailed balance has been employed to derive the following expression for calculating energy accommodation coefficient:$$a<\Theta^{2}(\gamma -1)/(\gamma +1)T_{a}T_{p},$$
where Θ is the Debye temperature. The calculations [36] showed that for Θ = 300 K and for the temperatures in gas *T_g_* = 3000 K and in condensed phase *T_s_* = 300 K, the accommodation coefficient α = 1/400 (monatomic gas) and α = 1/600 (diatomic gas). Thus, for typical values of metal particles parameters in flame the estimated value of α is of the order of 0.001, which is substantially (anomalously) lower of known in literature values [37]. Such an extremely low accommodation coefficient value was used in [25] to fit properly the available experimental ignition delay data with calculation results of Al nanoparticles heat transfer in conditions of shock wave. This approach was also cited in reviews [15,17] justifying the statement [36] about nanoparticles’ “thermal isolation” in a free-molecular heat-transfer regime.

However, recent experimental and theoretical data do not support the statement about significant thermal isolation of nanoparticles in a gas environment. In detailed experiments [30,31] on size resolved high-temperature oxidation of Ti and Zr nanoparticles (20–150 nm size), it was revealed that use of a very low value of energy accommodation coefficient (0.005) does not provide good correlation with several listed models: shrinking core-kinetic (with a kinetic controlled reaction), shrinking core-diffusion through ash layer (with fast kinetics and diffusion rate limiting), and Avrami–Erofeev nucleation model (with oxygen dissolution in the unreacted core). When treating energy accommodation coefficient as a best-fit free parameter the good result was achieved with the value α = 0.3. This is shown in Figure 11 for 40 nm Ti particle. Similar results were obtained for different sizes of Ti and Zr particles.

A sort of comprehensive theoretical analysis of determination of non-equilibrium energy accommodation coefficients for aluminum-inert gas systems is presented in [38]. Calculations of the coefficients for aluminum surface temperature of 300 K and gas (helium, argon, and xenon) temperatures within the range 1000–3000 K were performed in the framework of molecular dynamics approach. Additionally, based on density functional theory, the simulations of the gas–surface interaction potentials were accurately conducted, and the effects were determined by gas temperature and molecular weight on the accommodation coefficient. The calculated coefficients were found to be of the order of 0.1 (0.2–0.3 at 1000–3000 K argon temperature) and became essentially larger of the values predicted for similar conditions by Altman’s model [36]. Opposite of Altman’s model, the analysis [38] predicted weaker gas temperature dependency of the accommodation coefficient.

These estimates correlate well with classical representations [37,39] regarding dependencies of accommodation coefficient value on properties of gas and solid surface, and they do not depend (opposite of [25]) on assumptions made in the frameworks of ignition and combustion models. It can be mentioned that in the series of studies performed by a group with Prof. E.L. Dreizin the relatively large values of accommodation coefficient have been used to calculate the Al particle temperature history (α = 0.87 [40], α = 1 [10,41]). In [42], experimental dependency of ignition temperature on Al particle diameter [28] has been matched with theoretical calculations when using α = 0.5 value.

It was stressed in [10] that using a smaller-than-unity value of accommodation coefficient—not taking account of the finite rate of surface reactions—in calculations would lead to predicting an overestimated temperature of particles. To avoid this, it would be necessary to essentially reduce both the thermal accommodation coefficient and sticking coefficient (fraction of collisions of oxidizer molecules with the particle surface resulting in a chemical reaction). Namely, that was done in [25], where anomalously low values of α = 0.0035 and ϕ = 0.0009 were chosen to fit experimental data to calculations when using the chosen kinetic model. Obviously, use of such coefficient values could not justify the kinetic models employed in calculations, and future work is needed for quantifying the thermal accommodation and sticking coefficients. This may become important also for micron-sized particles, which are ignited and burned in a transition regime.

Despite the existence of numerous data on energy accommodation in different gas–solid systems, these could not be used in conditions that are of interest in combustion processes. Direct measurements of the accommodation coefficient are still very scarce. It should be noted that the data available in the literature about α-values mostly relate to pure substances. There exist discrepancies between the experimental data obtained under “surface science” conditions (ultrahigh vacuum, low coverage, low temperatures) and conditions of nanoparticles oxidation in practically important cases. The factors, such as admixtures on the evaporation surface or the chemical reactions between the vapor and components of gas environment, which proceed simultaneously with the evaporation, are believed to significantly affect the accommodation coefficient value. In addition, in the case of polyatomic gas molecules, it is necessary to make account of changing both translational and internal modes of energy of gas molecules transfer [37]. It is assumed in many cases that there is equilibrium between translational and internal freedom degrees of a polyatomic gas and thus a single accommodation coefficient can be employed. However, such an assumption is not well substantiated because the relaxation usually is faster for the translational degree of freedom.

To take account of that, it can be suggested using separate partial accommodation coefficients for the translational (α_tr_) and for internal (α_rot_) energy [37]. Their determination is a rather difficult technical task that takes special arrangement of the experiment. Such measurements were conducted in [43,44] with gilded tungsten wires of 8.3 μm diameter in a low-density wind tunnel. The initial pressure was about 2.7 Pa, stagnation pressure 1000 ÷ 2100 Pa, and stagnation temperature 300 K. The experiments were performed with monoatomic gases Ar and He and polyatomic ones H_2_, N_2_, CH_4_, and CO_2_. In the treatment of experiments with polyatomic gases, it was taken into account that, at the room stagnation temperature and wire temperature below 400 °C, only rotational freedom degrees are active. The values of accommodation coefficients were obtained by accurate numerical solving of the heat balance equation for wire of which the temperature was precisely measured with the use of the resistance method. There were obtained the values of α_tr_ comprising 0.29 (He) and 0.8 (Ar) as well as 0.67 (N_2_), 0.78 (CH_4_), and 0.9 (CO_2_). In addition, the values of α_rot_ were measured as 0.56 (N_2_), 0.73 (CH_4_), and CO_2_ (0.79). The data on α_tr_ are in a qualitative agreement with the data known from the literature. However, there are no similar data available in the literature regarding α_rot_. It is seen that the values of α_tr_ slightly exceed the values of α_rot_ and knowledge of both allows describing more precisely the heat exchange of gas with a solid surface. Most interesting experimental result of [43,44] is that for H_2_, the value of α_rot_ was found to be negative (−0.15). This contradicts to traditional definition of internal energy accommodation coefficients and can be explained by partial conversion of rotational energy into kinetic energy of scattered gas molecules. Note that in the literature one can find the discussion on the limits of accommodation coefficient magnitude [45,46]. It is stated there that these limits are 0 ≤ α ≤ 2, and correct description of gas–surface heat exchange has to include an account of translational, rotational, and vibrational regimes for gas molecules.

## 5. Determination of the Ignition and Extinction Time Instants

The data on metal particles ignition delay and burning time are necessary for both practical applications and establishing and substantiating proper ignition and combustion mechanisms. In the case of nanoparticles, there arise specific problems due to their diminishing sizes, which do not allow for making direct photography of individual particles. Therefore, temporal characteristics of nanoparticles motion and reaction are recorded mainly by measuring the length of tracks or the luminosity duration of reacting material.

One of the first detailed investigations [18] of Al nanoparticles ignition under shock wave conditions discussed different methods of determining the burning time. These methods include the constant intensity threshold, the full-width at half-maximum (FWHM), and the percent integrated light intensity approaches. As the light-intensity signals often have several peaks, especially at low oxidizer concentrations (or weak oxidizers use), it is believed that the method of percent-area may provide the most unambiguous determination of burn time. An example of application of this method is presented in Figure 12 for Al nanoparticles ignited in a shock wave in the mixture of CO_2_/N_2_ (50/50).

The reaction (burning) time is denoted in Figure 12 as the time corresponding period between points 10 and 90% of the total integrated intensity signal. The same procedure was also used to determine burning time in experiments with the mixtures of O_2_ with N_2_, where a single peak of light-intensity signal has been recorded. Note that the recorded light signals contained a relatively constant background emission signal, which must be deleted from the total record. This fitting procedure together with the procedure of choosing the characteristic points on the light intensity trace lead to a random uncertainty of approximately 20% in the value of burn time. The experiments were performed at pressures of 8 and 32 atm, with the temperatures behind the reflected shock being varied in the range of 1200–2100 K. As a result of the Arrhenius fitting for 80 nm Al particles (O_2_/N_2_ = 1:1), the activation energies were obtained, which comprised 71.6 ± 24.6 kcal/mol at 8 atm and 50.6 ± 15.2 kcal/mol at 32 atm. Additionally, it was revealed that when the method of FWHM was used for recording the burning time, the obtained energies of activation became about 50% higher than in the case of measuring with the 10–90% area method.

In later research, when working with nano- and micron-sized Al particles [20], it was suggested to use for detecting the burning time the light emission from the burning particles at wavelength 486 nm corresponding to AlO, which is intermediate of aluminum combustion. Its appearance characterizes the ignition, while disappearance stands for extinction, correspondingly. The burning time was measured with use of 486 nm records employing the 10–90% percent-area method. It has established very little AlO emission for nano-particle combustion that indicates occurring reactions mainly in the condensed phase. Therefore, in subsequent shock tube research with nanoAl, only the visible light intensity has been recorded [25,47].

Interesting information about burning time and behavior of nanoAl particles was recently reported in [48]. In this work the particles were injected at atmospheric pressure along the centerline of the burner directly into the products of the flat diffusion flame (CH_4_/O_2_/N_2_). The experiments were conducted with commercial aluminum nanoparticles (ALEX) of 50 nm primary size and with a 1.6 μm mean size electrospray generated mesoparticles composed of original ALEX particles and nitrocellulose. For tracking the burning particles, a high-speed video camera was used focused directly at the burner centerline. Consequently, the burning time was determined by recording and counting the number of individual particles frames. It was established in detailed experiments that the burning time of mesoparticles is practically one order of magnitude shorter than the burning time of ALEX particles. According to scheme presented in Figure 13, this is caused by the fact that the primary ALEX particles already exist in the form of hard agglomerates with intraparticle necking, which may sinter immediately under heating that results in the formation of much larger particles participating in the real combustion process. In contrast, assembled with nitrocellulose mesoparticles burn as fast as the original smallest hard aggregates in the nanopowder. These results correlate with experimental findings [19], which revealed that for Al nanoparticles burning time is about 500 μs in a shock tube at 8 atm pressure, 1400 K, and a 50% O_2_ environment. It is believed that the shock-induced breakup of the large agglomerates and thus the shock tube results correspond to the combustion of the smallest nanoAl aggregates in the aerosol.

The pioneering study of the ignition and combustion of single crystalline boron particles was conducted more than 50 years ago [49]. In that work, the particles of boron with average diameters of 34.5 and 44.2 μm were introduced at atmospheric pressure in the post-flame (CO or propane with O_2_) zone of a flat-flame burner. The experimental procedure of particle combustion studies has been described previously in earlier work [50]. To produce introduction of particles, a 250 micron bore hypodermic needle is mounted axially terminating at the upper surface of the burner disc. Initial particle velocity comprises few cm/s and the burnt-gas velocity is of the order of 1 m/s. The particle’s travel distances on photographs are converted to residence times on the basis of known gas flow velocity, and a certain correction has to be made to account for initial acceleratory period. The estimation of the correction coefficient [50] for typical parameters of gas flow showed substantial correction value (about 1.5–2.0). Like in several previous studies, it was established [49] that the combustion of boron particles exhibits two successive stages. The first combustion stage is associated with the burning of boron particles initially covered with an oxide layer, while second combustion stage is associated with fully fledged combustion of the bare boron particle. Such representation of boron particles’ combustion mechanism was confirmed and examined in detail later in numerous research dealing with micron- and nano-sized boron particles [26,51]. In [26], the burning time of 1 μm amorphous boron particle was determined, similar to [49], by dividing the length of the burning streak by the average particle velocity.

Another approach to measure the ignition and burning time of crystalline boron particles at pressures 30–150 atm was presented in [52]. The particles of 24 μm mean size were injected into a constant volume combustion bomb filled with the combustion products of N_2_ diluted hydrogen/oxygen mixtures (O_2_ excess concentrations 5–20% and temperatures 2440–2830 K). Note that in these conditions, only single-stage combustion has been observed. Experimental diagnostics consisted of dynamic pressure measurement, particle injection timing, and optical emission detection. The latter was done at BO_2_ molecular band (interference filter at 546.1 ± 4 nm), corresponding to intermediary gas-phase species in B combustion. Figure 14 shows graphical procedure for determining the particle combustion characteristic time.

Following a methodology similar to that described earlier in [53], the ignition delay is defined as the time period from the instant of particle injection to the point of ignition indicated as the half of emission maximum value. The burning time is defined as the time from ignition to the moment when the filtered photodiode output signal diminishes to half maximum value. For comparison, a signal of pure B_2_O_3_ is shown indicating that the oxide emission is significantly lower even though BO_2_(*g*) molecules are produced by the B_2_O_3_ particles injected in a hot gas in separate experiments. Taking into account previously determined size distribution of particles, it was suggested to choose the half height of the net particle signal as the ignition point because this moment corresponds to mean-sized particles emittance, while the time of first light corresponds to the signal from smallest size particles comprising a small percent of the total mass. The same considerations were taken into account when defining the burning time. Further, determined in this way, the characteristic times were compared to predictions from two ignition models. In particular, excellent agreement of experimental burning times with predictions by a detailed-chemistry Princeton/Aerodyne model in 20% O_2_ mixtures and discrepancy (over prediction) for lower *X*_O2_ mixtures was established. That discrepancy is believed to arise from the experimental interpretation of the instant when particle combustion is completed.

When dealing with nano-sized B particles, the problem arises of how to distinguish correctly the first and second combustion stages. To solve this problem, it was suggested [9] to analyze high-speed video images of glowing particles made with and without interference filter, centered at 546 nm. Earlier, the emission on this band was not detected during the first stage of combustion [54]. Therefore, the filtered images are assumed to provide the spatial location of second stage in B combustion, whereas unfiltered images indicate starting location for the first combustion stage. An example of unfiltered image of SB99 (72.2% pure B) nanopowder combustion is shown in Figure 15. This powder has a primary particles size about 62 nm and aggregates size 200–320 nm.

Note that in [9], the particles were injected perpendicular to the burner flow and traversed the diameter of the burner. To obtain the velocity profiles of the injected particles in the post-flame region, the detailed PIV particle image velocimetry experiments were conducted, which allowed making reasonable estimates of ignition and burning times. It is seen in Figure 15 that there are two distinct parts of image luminosity where the first one (yellow) corresponds to ignition and the second (white glow) to fully fledged combustion. The qualitative treatment of this two-dimensional picture is rather difficult task, and it was suggested to convert it into one-dimensional by summing the columns (or the *y*-dimensions) of the images with subsequent constructing the profile in the *x*-direction. Then an area-based method was employed in order to provide the reliable determination of the burning time [55]. As is shown in Figure 16, the burning time was detected as occupying 95% of the area of the original intensity profile, with the blue line representing the original profile of filtrated image and the red line representing 95% of the total area. Subsequently, the burning time t_2_ defined in this way was used to compare with the measurements of different authors (see Figure 2, present paper) and to derive the apparent energy activation from Arrhenius burning law correlation in the form of 1/t_2_~const exp(−E/RT).

A similar approach was used in [30,31] for determining characteristic times of high-temperature reactions of nano-sized (20–150 nm) titanium and zirconium particles. Nanoparticles were produced by a metal target pulsed laser ablation and then the size was selected with use of a differential mobility analyzer. To evaluate the total burning time, a high-speed camera recorded the particles luminosity streaks with an exposure greatly exceeding the particle burning time. In this case, the whole streak length was used to determine the burning time on the basis of known gas velocity. The experiments were conducted with highly diluted aerosol of the size-selected metal particles, which were injected into a high temperature (1700–2500 K) oxidizing zone produced by a laminar CH_4_/O_2_/N_2_ diffusion flame. It might be noted that for the size-selected burning times, the data scatter was noticeably small (2~9%) indicating narrow range of burning times for the particles of a given size. This serves an advantage of developed approach and strengthens fidelity of experimental data.

To illustrate a variety of existing methods to determine metal particles’ burning time, one can consider fresh information presented in [56]. Fuel-rich boron B composites containing bismuth fluoride (B·BiF_3_) and bismuth (B·Bi) were prepared by arrested reactive milling and then characterized using field emission scanning electron microscope to obtain the particles’ size distributions. The aerosolized particles were directed into the focal zone of CO_2_ laser producing particles ignition at room temperature in air at atmospheric pressure. A photomultiplier tube (PMT) was used to record a light emission of ignited particles. The examples of the PMT records for composites with different content of boron (90–69%) are presented in Figure 17. The instant of ignition was determined from high-speed videography and the burning time was defined, as it was done previously in [57], as the period of time over which the PMT signal exceeds 10% of the peak maximum.

For obtaining the effect of particle size on burning time, the correlations between statistical distributions of the particle sizes and measured burning times were examined. Figure 18 presents the results obtained in the form of burning time–particle size diagrams for B·BiF_3_ and B·Bi composite powders and for pure boron [57] and 50B·50BiF_3_ samples [58], respectively.

## 6. Laser Heating of Nanoparticles

When describing the temperature history of metal particles under irradiation, the issue of radiation absorption has to be carefully discussed. The matter is that the radiation absorption efficiency does depend on the particle size, and this should be taken into account in the case of fine metal particles. Following [40], the laser energy delivered to the particle can be calculated as *q*_Laser_ = 0.25π *D*^2^*η* (*λ*, *D*, *n_i_*) *I_rad_*, where *η* is the particle laser absorption efficiency, which depends on the laser wavelength *λ*, *D* the particle diameter, *n_i_* the material complex refractive index; and *I_rad_* is the laser power density.

The energy absorption efficiency *η* is described in detail for spherical metal particles in [59]. Based on this approach, the absorption efficiency was calculated in [40] for the temperatures below the Al melting point using Mie’s scattering theory. The calculations revealed that for Al the absorption efficiency peak is realized at 3.37 μm particle diameter.

Similar calculations performed for other metals revealed that for a given laser wavelength (10.6 μm) the absorption efficiency estimated on the basis of Mie’s scattering theory has a maximum at approximately the same particle size (*D =* 3.37 μm). It can be supposed that due to the particle size selective heating, the particles corresponding to the peak absorption efficiency will be ignited mostly under laser irradiation. Later the model was modified to take into account the effect of melting [60] on the absorption efficiency. It was found that the particle absorption efficiency experiences a jump on melting, which is caused by abrupt density change.

The results of the calculations clearly demonstrate that the absorption efficiency depends on the particle size and its temperature. More information can be found in specialized literature sources. In common, the particle radiation absorptivity is characterized [59] by the absorption cross section, *C_a_*, which is calculated as the total amount of absorbed energy divided by the incident radiation intensity. Further, the efficiency to absorb radiation can be characterized by absorption efficiency of the particle based on its cross-sectional area, *η* = *C_a_*/(π*D*^2^/4), and the volumetric absorption efficiency based on its volume, *η_v_* = 6*C_α_*/π*D*^3^.

Detailed calculations for different metals showed that the absorption efficiency essentially depends on the temperature. It increases by about four, three, and two times for Ni, Al, and Cr particles when the temperature increases from 300 K to 900 K. Thus, the particles’ preheating can effectively enhance absorption of radiation in metallic particles.

In accordance with calculations, the micron- and submicron-size particles can absorb radiation more efficiently than larger particles because of strong diffraction effect at the particle surface. For nanometer particles, radiation absorption becomes less efficient. For vanishingly small particles (e.g., d < 10 nm), *η_v_* approaches a constant, which is the Rayleigh limit of light scattering by small particles. In case of metallic particles, the maximum *η_v_* is by about two orders of magnitude larger than the Rayleigh-limiting value.

From the point of view of practical applications, the knowledge of fine particles’ radiation absorption efficiency is important for estimation of ignition delays of energetic materials containing inclusions of highly absorbing substances. In particular, it may have sense in the case of laser heating of explosives seeded by nanoparticles of metal. Based on the Mie theory, the calculations were conducted [61] for inclusions of in total 12 different metals in the matrices of silver azide, lead azide, and pentaerythrite (PETN). The wavelength of radiation was 1064 nm and particle sizes varied in the range 0–600 nm. Figure 19 presents calculated dependencies of the radiation absorption efficiency on the radius *r* of inclusions of silver (*n_i_* = 0.15 − 6.0*i*) in a silver azide matrix (*n* = 2), lead (*n_i_* = 1.416 − 5.742*i*) in a lead azide matrix (*n* = 1.85), and lead and aluminum (*n_i_* = 0.978 *−* 8.030*i*) in a PETN matrix (*n* = 1.55). Here, *n_i_* is the complex refractive index of the metal.

It is seen from Figure 19 that silver inclusions in a silver azide matrix absorb radiation ineffectively. The maximum absorption efficiency *η* equals ca. 0.204, whereas for lead in a lead azide matrix, it comprises ca. 1.18. Small *η* value in the case of silver inclusions is related to the pronounced metal properties of silver and corresponds to low value of the real part of refractive index and high value of its imaginary part. Correspondingly, the larger *η* value in the case of lead inclusion is due to its less-pronounced metallic property. Aluminum demonstrates intermediate *η* values. Thus, when calculating the heating history of the inclusion, it is necessary to take into account the nature of the metal and use corresponding absorption efficiency.

An example of such calculations is presented in Figure 20. The calculations were conducted for a very short duration of laser pulse (30 ns) when the heat exchange with matrix and the effect of its chemical reaction can be ignored. The comparison is shown between the time history of heating with the account of real dependency radiation absorption on inclusion radius and in the case when it is assuming *η* = 1. It is evident that in the case of a “weak metal” (Pb) the maximum heating temperature is essentially lower when taking into account the real value of its radiation absorptivity.

## 7. Concluding Remarks

As is mentioned in the Introduction, the metal nanoparticles have a wide list of applications in various explosive and propulsion systems, including development of effective modifiers of the burning rate of solid propellants, advanced components of pyrotechnic compositions, etc. The research in that direction is conducted intensively in USA, Russia, China, and other countries. Different aspects of such applications are discussed in the books published in Elsevier in 2016 [62] and 2019 [63]. In particular, use of metal nanoparticles has allowed obtaining great burning rates in MoO_3_–Al nanothermites, which reach 790 m/s and provide perspectives for developing microscale rocket propellants [64]. Nanostructured energetic materials such as Al/Ni or Al/CuO can be used for fabrication of igniting bridges to facilitate ignition process with augmented output energy and igniting ability that allows initiation of high explosives [65].

An important advantage of studying the nanoparticles’ performance is that it allows exploring the intrinsic mechanism of heterogeneous reactions. For example, recently, some detailed studies [66] revealed that, in nano-Al-based thermite reactions, the major part of heat release is contributed by a condensed phase mechanism. Obviously, these studies take development of the proper approaches and techniques.

The present paper focuses attention on the analysis of specific features of studying the metal particles ignition and combustion, because the literature data contain sometimes ambiguous information on the characteristics of these processes that do not allow making substantiated statements about their mechanism and predicting reliably its temporal behavior. First of all, there is no possibility to study the reaction behavior of a single nanoparticle due to technical problems of direct recording (photography) parameters of very fine objects and due to extremely fast [13] sintering of small-size particles leading to formation of the hard aggregates. Even in the case of advanced techniques of laser ablation or arc discharge [29], the data obtained correspond to relatively small-size aggregates of original nanoparticles. Nevertheless, the advantage of such techniques is that they allow us to indicate the real size of an aggregate in contrast to the earlier techniques with injection of finite size distribution powders into the shock tube or post-flame zone of the burner.

Another problem is the correct determination of ignition and burning times. It is evident that various research groups use different methods to detect ignition and extinction instances (10% of maximum luminosity signal; 90 or 95% of integrated signal). Moreover, in most of the literature sources, it is assumed that particles immediately enter into a hot gas zone with ignorance of the effects of carrying gas flow on the temperature and the oxygen concentration in environmental gas. Obviously, the account of these effects may give more substantiate interpretation of observed experimental facts. For instance, it was recently explained [16] that the enhancement of the burning rate of boron nanoparticles in a wet hot gas zone (with water vapors) can occur due to changing the heat-exchange intensity (increase in the Nusselt number due to increase in the velocity of gas exhaust from the burner). Note that no increase in the compact boron samples burning rate in wet gas environment was experimentally observed in [67]. It was also underlined in [16] that there exist intrinsic difficulties in detecting the final stage of large particles combustion because the amount of energy released at the particle trace where the particle size is below 3 μm becomes smaller at least by a factor of 1000 and an equipment tuned to detect signals of large particles may fail to capture such low-intensity signal.

It is necessary to mention here the issue of correct interpretation of experimental burning rate vs. time dependencies related to correct using the values of the energy accommodation coefficient. It was proposed recently [68] to employ for Al nanoparticles logarithmic burning law *t_b_~lnD* instead of commonly used power law *t_b_*~*D*^0.3^. The last law simply represents an experimental fit to data, and the power value 0.3 does not have physical meaning. The logarithmic law is derived on the basis of a simplified model based on the energy balance on a burning metal nanoparticle surface. This model utilizes a small value of energy accommodation coefficient previously used to explain shock tube experimental data [25]. The original idea on two different regimes of metal nanoparticles’ combustion, which may exist depending on the value of energy accommodation coefficient, is argued in a fresh paper [69]. The illustration of a new idea is presented in Figure 21. Red and green triangles represent burning times of 4 μm Al particles in two combustion regimes. Solid lines correspond to available literature data, while dashed line corresponds to a hypothetical regime.

It is assumed that if a small size Al particle were burning with a high energy accommodation coefficient, the burning time-particle diameter dependency would be much stronger than the experimentally observed one. The background for such an assumption can be found in experiments with pre-stressed via fast cooling the heated Al particles [70]. It was established that the shell-core delamination associated with the fast cooling the Al particles facilitates diffusion reactions. This results in reducing a burning time. Thus, the altering of the shell-core particles’ mechanical properties leads to altering of their reaction mechanism. The considerations related to electron energy transfer on the interface between a gas and metal and its contribution to the energy accommodation coefficient are discussed in [71]. Numerical calculations show that the coefficient magnitude decrease by a couple of orders of magnitude is possible. However, the approach used has a semi-empirical nature, and as is mentioned in [68], it does not pretend to be a part of comprehensive theory of the nanoparticle combustion but can be employed for the development of its objective background.

Finally, global problems exist regarding how to compare correctly and generalize the data on the ignition and burning time vs. particle size dependencies, which are obtained with use of different techniques of particles’ injection into a hot gas zone and use of different methods of reaction time measurement. Besides, reported data are obtained for particles of different metal purity and different states of oxide layer. Obviously, at present, such a comparison can be made mainly on a qualitative level, and it is necessary to characterize in detail all relevant parameters of a particle’s material quality and measurement techniques. In addition, it is necessary to continue developing the advanced approaches for obtaining and characterization of narrow fractions of nanoparticles and for detailed description of the particles’ dynamic behavior in hot gas environment.

## Figures and Tables

**Figure 1 nanomaterials-10-02008-f001:**
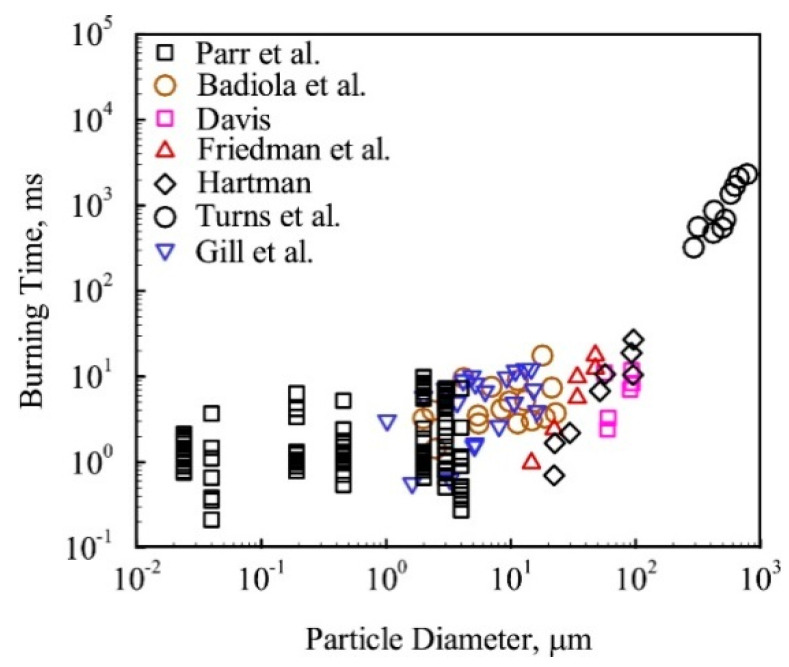
Burning time vs. aluminum particle size exhibiting weak size effect at nano-scales. Reproduced from [5], with permission from Prog. Energy Combust. Sci., 2017.

**Figure 2 nanomaterials-10-02008-f002:**
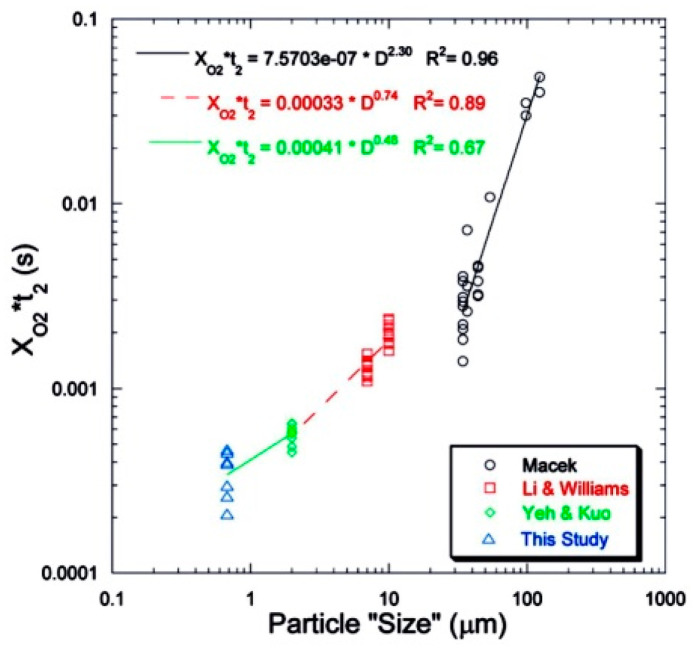
Comparison of the full-fledged combustion times of B particles (Macek [6], Li [7], Yeh [8], this study [9]). Reproduced from [9], with permission from Elsevier, 2009.

**Figure 3 nanomaterials-10-02008-f003:**
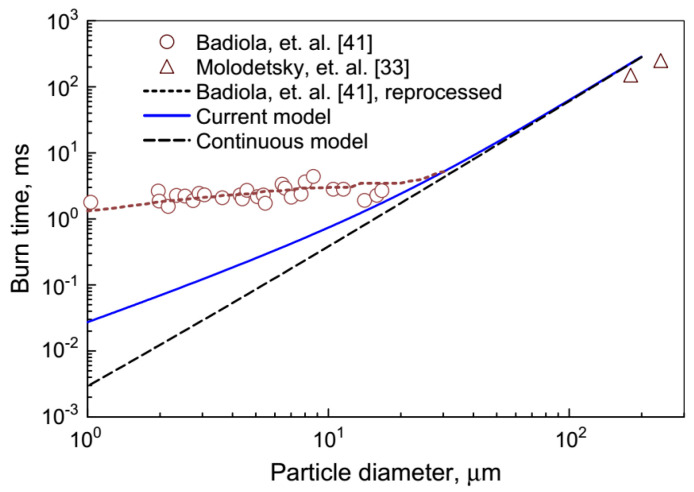
Dependency of the burning time on Zr particles size (Badiola [10], Molodetsky [11]). Reproduced from [12], with permission from Elsevier, 2013.

**Figure 4 nanomaterials-10-02008-f004:**
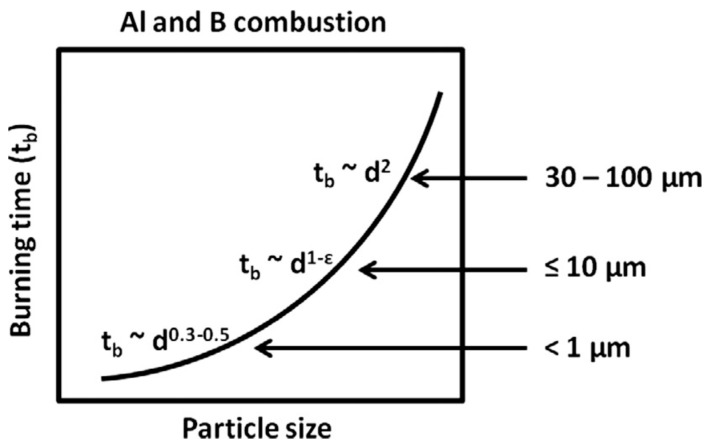
Conceptual figure showing determined experimentally for Al and B the burning time on size dependency. Reproduced from [13], with permission from Elsevier, 2014.

**Figure 5 nanomaterials-10-02008-f005:**
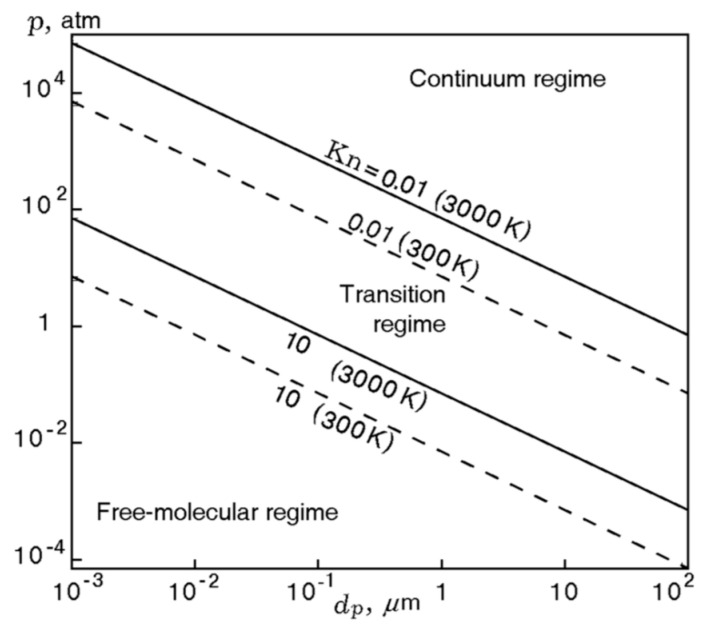
Boundaries between the continuum and free-molecular regimes as a function of particle size at different pressures and temperatures 300 and 3000 K. Reproduced from [14], with permission from Elsevier, 2015.

**Figure 6 nanomaterials-10-02008-f006:**
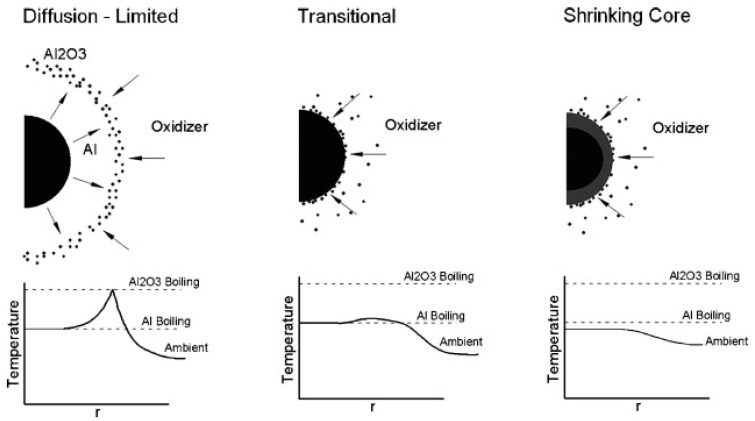
Schematic of the flame structures presenting in aluminum combustion. Reproduced from [19], with permission from Prog. Energy Combust. Sci., 2017.

**Figure 7 nanomaterials-10-02008-f007:**
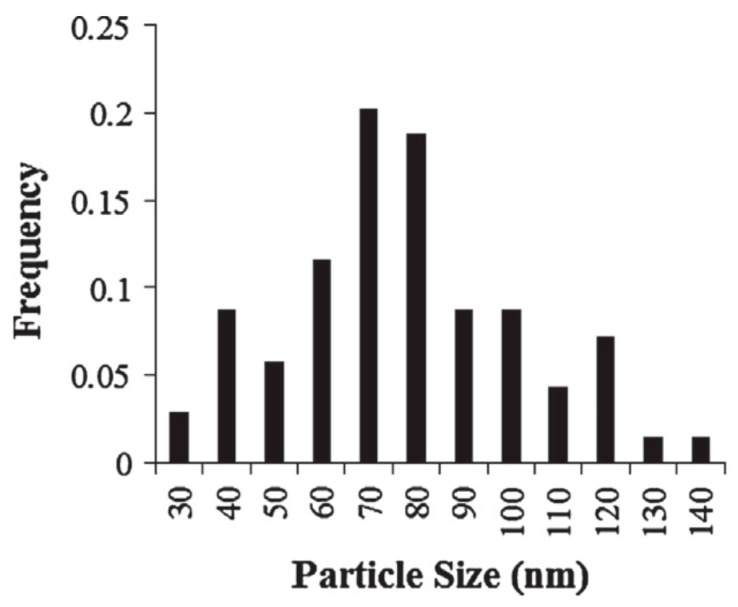
Particle size distribution (50 nm nominal size, SkySpring Nanomaterials). Reproduced from [25], with permission from Elsevier, 2014.

**Figure 8 nanomaterials-10-02008-f008:**
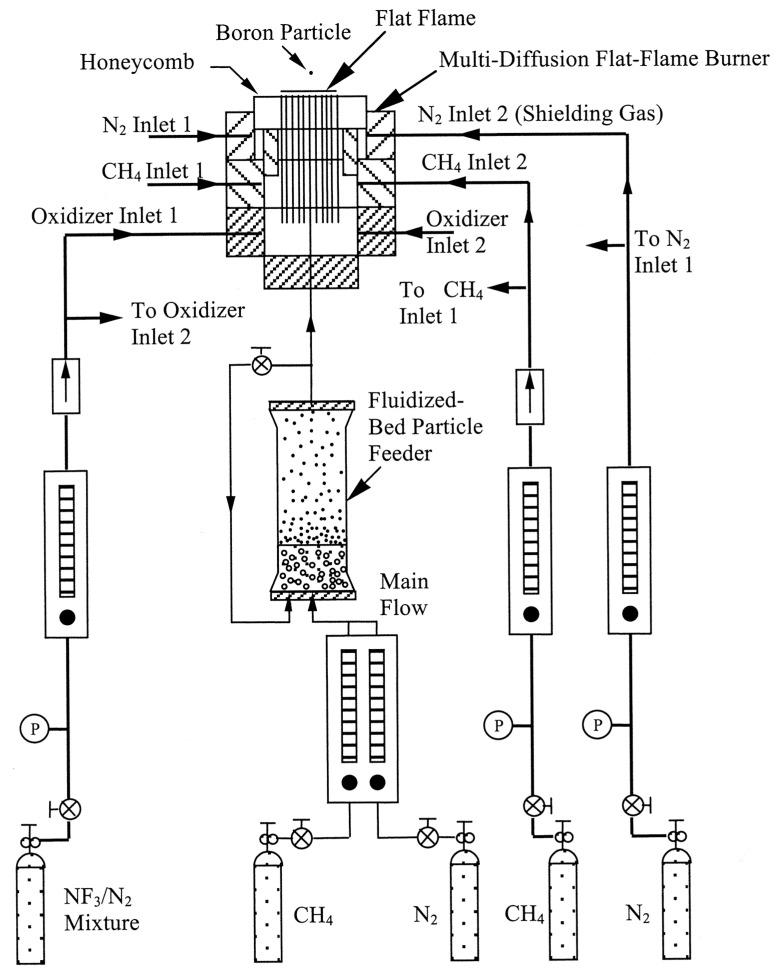
Schematic of used the burner setup and gas supply system. Reproduced from [26], with permission from Elsevier, 2001.

**Figure 9 nanomaterials-10-02008-f009:**
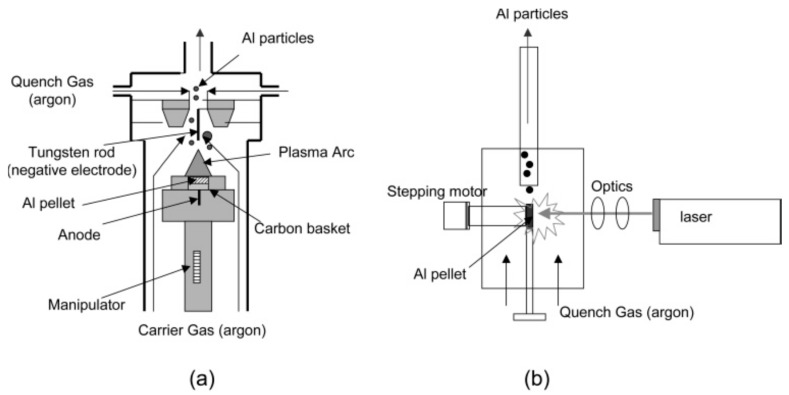
Methods used for generating nanoparticles of Al: (**a**) through DC arc discharge, (**b**) via laser ablation. Reprinted from [29] with permission from J. Phys. Chem. B. Copyright 2005 American Chemical Society.

**Figure 10 nanomaterials-10-02008-f010:**
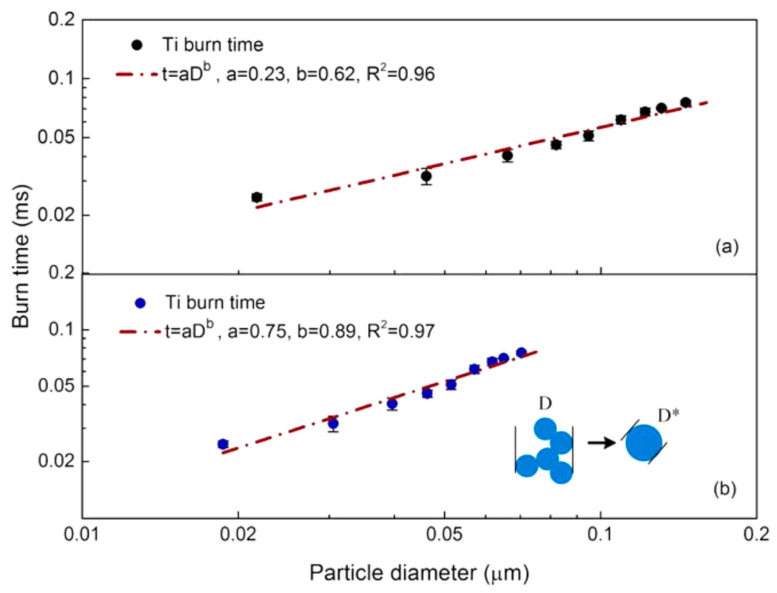
Ti particles burning time vs. particle size diagrams: (**a**) based on the peak DMA selected particle size, (**b**) based on the TEM measured diameter after sintering. Reprinted from [30] with permission from J. Phys. Chem. A. Copyright 2015 American Chemical Society.

**Figure 11 nanomaterials-10-02008-f011:**
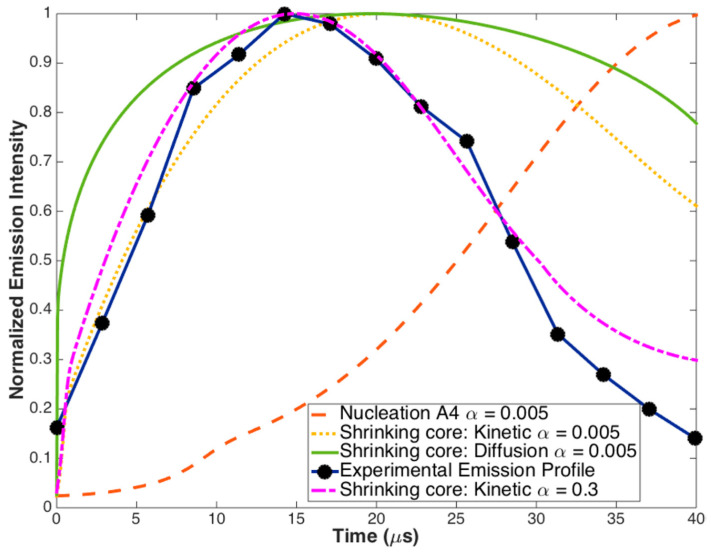
Model simulations for 40 nm Ti particle with α = 0.005 and α = 0.3. Reprinted from [30] with permission from J. Phys. Chem. A. Copyright 2015 American Chemical Society.

**Figure 12 nanomaterials-10-02008-f012:**
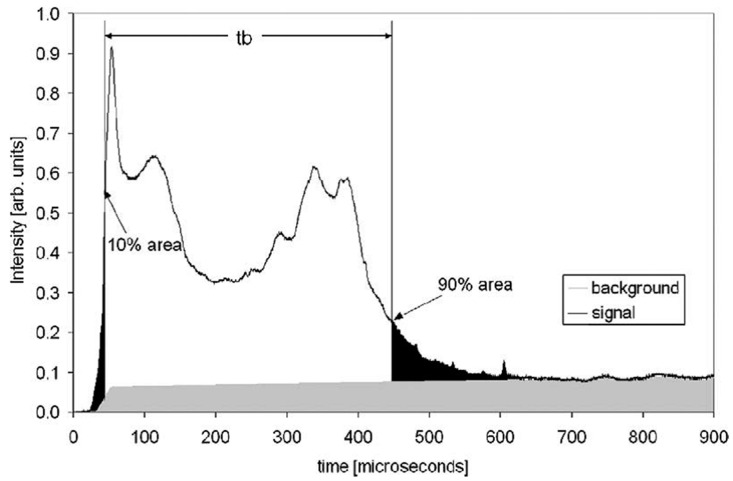
Visible light intensity vs. time traces for nanoaluminum in 50% CO_2_ and 50% N_2_ at 1760 K and 32.4 atm. Reproduced from [18], with permission from Elsevier, 2006.

**Figure 13 nanomaterials-10-02008-f013:**
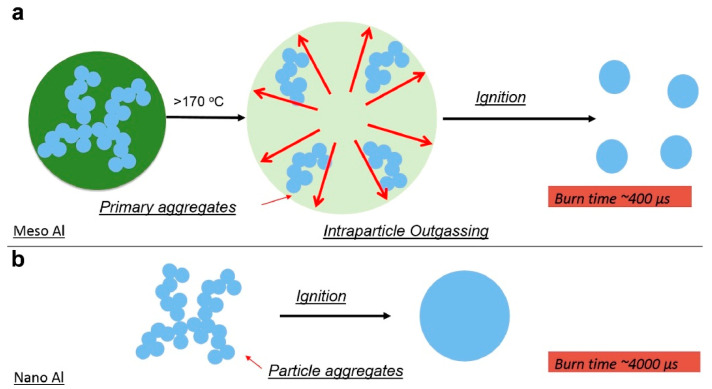
Overall pictures of the events occurring in the combustion of: (**a**) Al/NC mesoparticles, (**b**) Al nanoparticles ALEX. Reproduced from [48], with permission from Elsevier, 2016.

**Figure 14 nanomaterials-10-02008-f014:**
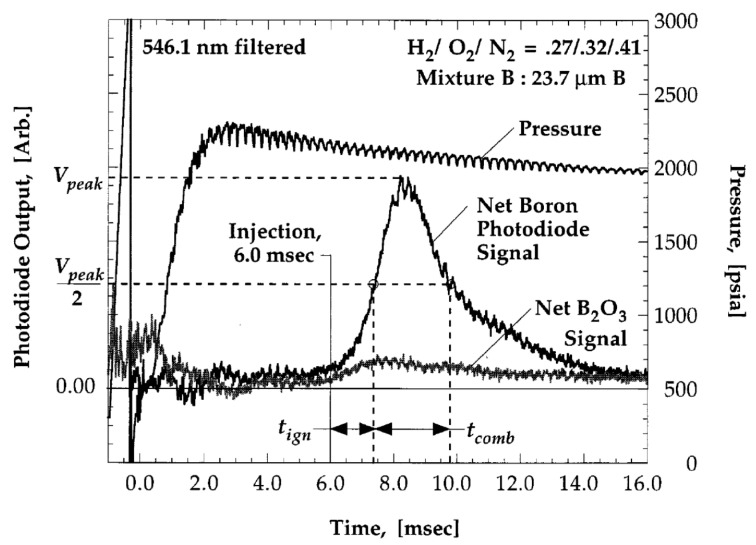
Filtered at 546.1 nm signal of B particle burning in gas mixture (H_2_/O_2_/N_2_ = 27/32/41). Reproduced from [52], with permission from Elsevier, 1999.

**Figure 15 nanomaterials-10-02008-f015:**
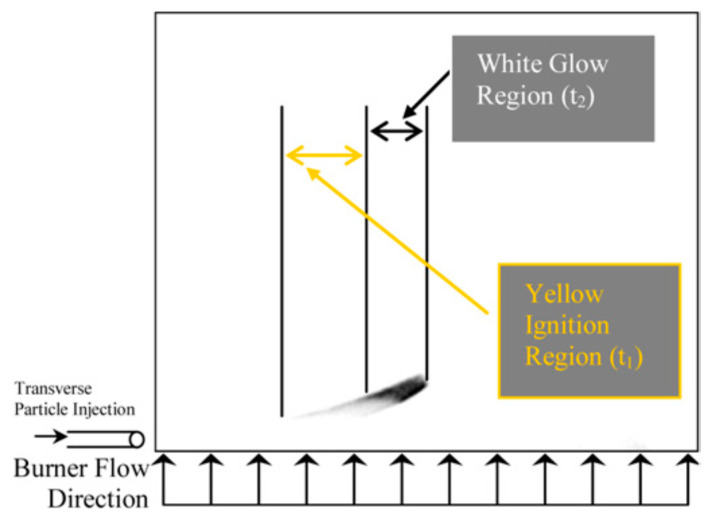
Unfiltered image of SB99 nanopowder combustion. Reproduced from [9], with permission from Elsevier, 2009.

**Figure 16 nanomaterials-10-02008-f016:**
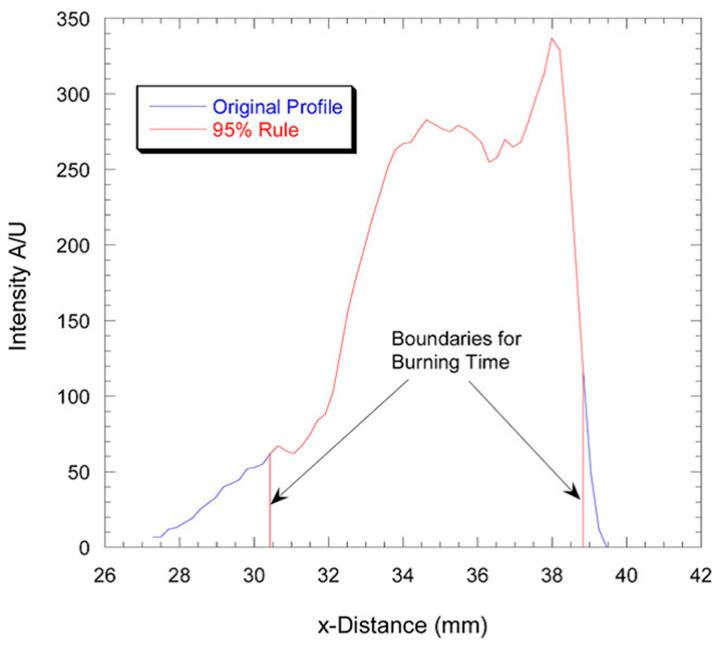
Illustration of t_2_ burning time determination (X_O2_ = 0.2, T = 1808 K). Reproduced from [9], with permission from Elsevier, 2009.

**Figure 17 nanomaterials-10-02008-f017:**
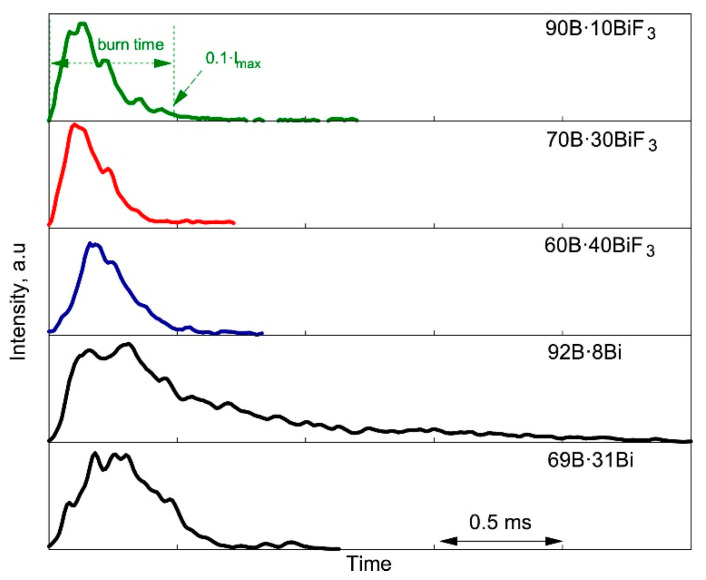
Characteristic emission records of the burning in air particles of B·BiF_3_ and B·Bi composites. Reproduced from [56], with permission from Elsevier, 2020.

**Figure 18 nanomaterials-10-02008-f018:**
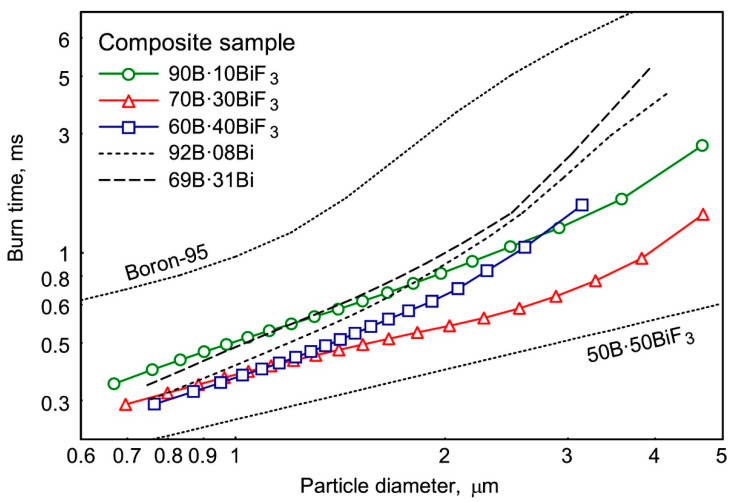
The burning time–particle size correlations for B·BiF_3_ and B·Bi composite powders burning in air (Boron-95 [57], 50B·50BiF_3_ [58]). Reproduced from [56], with permission from Elsevier, 2020.

**Figure 19 nanomaterials-10-02008-f019:**
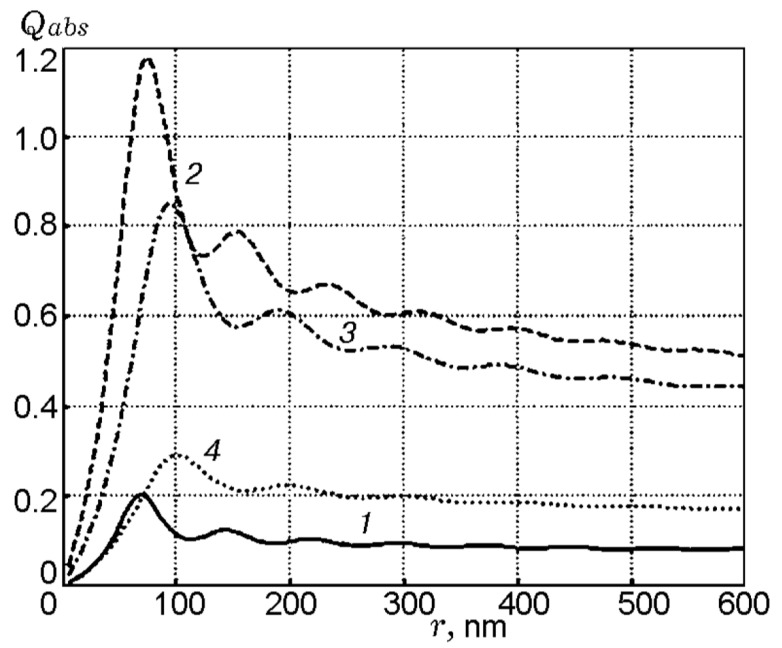
Efficiency of radiation absorption (*λ* = 1064 nm) by metallic inclusions in a matrix of energetic materials vs. radius of inclusions: Ag in AgN_3_ matrix (*1*), Pb in PbN_6_ matrix (*2*), and Pb and Al in PETN matrix (*3* and *4*). Reproduced from [61], with permission from Combust., Explos., Shock Waves, 2012.

**Figure 20 nanomaterials-10-02008-f020:**
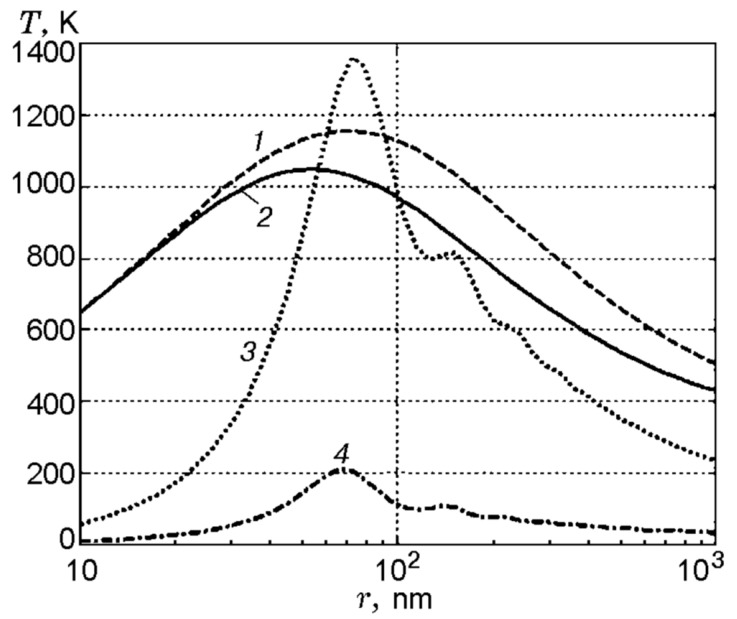
Maximum heating temperature of metal nanoparticles in an inert matrix having thermophysical properties of PbN_6_ (laser pulse length 30 ns and energy density of 50 mJ/cm^2^): curve *1* stands for Ag (*η* = 1) and curve *2* to Pb (*η* = 1); curves *3* and *4* refer to Ag and Pb with account of the dependency *η*(*r*). Reproduced from [61], with permission from Combust., Explos., Shock Waves, 2012.

**Figure 21 nanomaterials-10-02008-f021:**
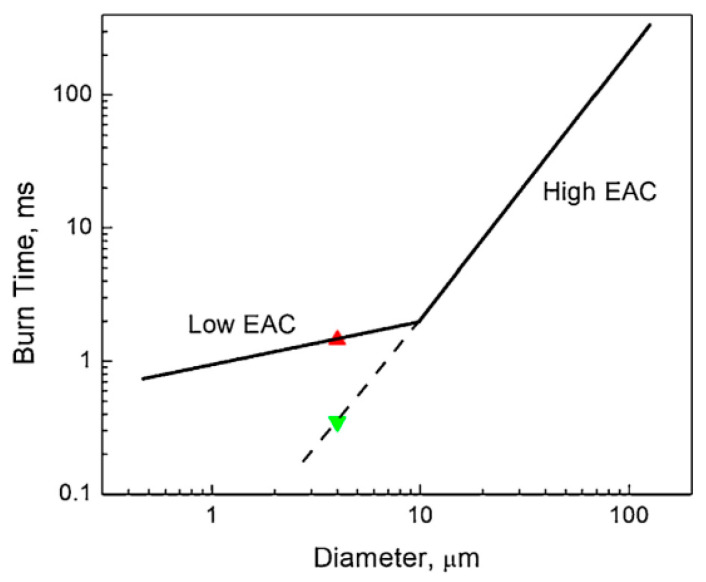
Burning time vs. Al particle diameter dependency. Reproduced from [69], with permission from Elsevier, 2020.

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
