# Peer review of "Review of Experimental Methods for Measuring the Ignition and Combustion Characteristics of Metal Nanoparticles"

_nanomaterials, 2020, doi:10.3390/nano10102008_

Round 1

Reviewer 1 Report

The authors provided a very complete review on the methods for the measurements of the ignition and combustion characteristics of metal nanoparticles, which is an interesting and not fully deepened yet topic. 

The paper is interesting and well-written, and clearly presents an an exhaustive review of the work done in this field, which could represent a nice add to the literature. However, I believe that English can be improved and some typos have to be corrected before the manuscript can be considered suitable for publication:

Author Response

Response to Reviewer 1 Comments

Point 1: The paper is interesting and well-written, and clearly presents an an exhaustive review of the work done in this field, which could represent a nice add to the literature. However, I believe that English can be improved and some typos have to be corrected before the manuscript can be considered suitable for publication:

Response 1: The author re-examined the text and did his best trying to improve it.

Reviewer 2 Report

I find this work valuable since it is an analysis that help the reader to assess the quality of published experimental studies and methods. Metal combustion is an emerging field of reseach and reviews such as this are important.

I have no major comments but note that some figures does not reproduce well on my printout, in particular figure 2, 5 and 16.

In section 3 there are both spaces between sections and indented first lines in each paragraph, this need to be corrected.

Author Response

Point 1: I have no major comments but note that some figures does not reproduce well on my printout, in particular figure 2, 5 and 16.

Response 1: The author re-examined the text and improved reproduced figures quality.

Point 2: In section 3 there are both spaces between sections and indented first lines in each paragraph, this need to be corrected.

Response 2: The author did not find places in section 3 to improve.

Reviewer 3 Report

I think the topic of this manuscript is important and it is well-written. It could be accepted by nanomaterials as it is. 

Author Response

Point 1: It could be accepted by nanomaterials as it is.

Response 1: The author sincerely thanks the reviewer.

Reviewer 4 Report

The authors summarized recent experimental methods for characterizing the ignition and combustion of metal particles, such as aluminum and boron. Also illustrated the difficulty of studying the combustion of single metal nanoparticles and the related experimental approaches towards this matter along with the summary on methods measuring the sizes of different metal nanoparticles. Different methods of determining ignition delay and burn times are also introduced and compared for the sake of the practical applications and theoretical simulations. Overall the authors did a clear and thorough summary of the recent progress of experimental methods for the combustion of metal nanoparticles. This article could provide great help for researchers who are in this field. Therefore, I recommend its publication on Nanomaterials after minor revision. In introduction, the authors must cite nanothermites synthesis as huge field of applications of metal nano particles (L41 p1) and cite major works and teams involved in nanothermites as well.

Author Response

Point 1: In introduction, the authors must cite nanothermites synthesis as huge field of applications of metal nano particles (L41 p1) and cite major works and teams involved in nanothermites as well.

Response 1: In order to avoid changing too much the references numeration the author added 15 lines (L680-694, p. 20) to the final part of the text and 5 new references (#62-65, p. 24) for underlining the application of metal nanoparticles in nanothermites.